# Genetic Susceptibility in Head and Neck Squamous Cell Carcinoma in a Spanish Population

**DOI:** 10.3390/cancers11040493

**Published:** 2019-04-07

**Authors:** Javier Fernández-Mateos, Raquel Seijas-Tamayo, Juan Carlos Adansa Klain, Miguel Pastor Borgoñón, Elisabeth Pérez-Ruiz, Ricard Mesía, Elvira del Barco, Carmen Salvador Coloma, Antonio Rueda Dominguez, Javier Caballero Daroqui, Encarnación Fernández Ruiz, Alberto Ocana, Rogelio González-Sarmiento, Juan Jesús Cruz-Hernández

**Affiliations:** 1Medical Oncology Service, University Hospital of Salamanca-IBSAL, 37007 Salamanca, Spain; javierfermat@gmail.com (J.F.-M.); raquel_seijas@usal.es (R.S.-T.); jcadansa@saludcastillayleon.es (J.C.A.K.); u93667@usal.es (E.d.B.); 2Biomedical Research Institute of Salamanca (IBSAL), SACYL-University of Salamanca-CSIC, 37007 Salamanca, Spain; 3Molecular Medicine Unit- IBSAL, Department of Medicine, University of Salamanca-CSIC, 37007 Salamanca, Spain; 4Institute of Molecular and Cellular Biology of Cancer (IBMCC), University of Salamanca-CSIC, Salamanca 37007, Spain; 5Medical Oncology Service, Hospital Universitario Politécnico La Fe, 46026 Valencia, Spain; pastor_migbor@gva.es (M.P.B.); carmen_salvador_@hotmail.com (C.S.C.); 6Division of Medical Oncology, Oncology department, Agencia Sanitaria Hospital Costa del Sol, 29603 de Marbella, Spain; eliperu@gmail.com; 7Medical Oncology Department, Universitat de Barcelona, IDIBELL, Institut Català d’Oncologia, L’Hospitalet de Llobregat, 08908 Barcelona, Spain; rmesia@iconcologia.net (R.M.); daroqui@gmail.com (J.C.D.); 8Medical Oncology Service, Hospital Regional Universitario de Málaga, 29010 Málaga, Spain; rueda.dominguez@gmail.com; 9Otolaryngology Agencia Sanitaria Hospital Costa del Sol, 29603 de Marbella, Spain; oncolandia@gmail.com; 10Hospital Clínico San Carlos, IdISSC, CIBERONC, 28040 Madrid, Spain; albertoo@sescam.jccm.es; 11Centro Regional de Investigaciones Biomédicas, Universidad de Castilla La Mancha, 13071 Albacete, Spain

**Keywords:** head and neck cancer, single nucleotide polymorphisms, Spanish population

## Abstract

Despite classical environmental risk factors like tobacco, alcohol or viral infection, not all individuals develop head and neck cancer. Therefore, identification of the genetic susceptibility produced by single nucleotide polymorphisms (SNPs) is an important task. A total of 296 human papillomavirus negative head and neck cancer (HNC) patients (126 laryngeal, 100 pharyngeal and 70 oral cavity) were included in the study, involving 29 candidate SNPs in genes within important carcinogenic pathways (oncogenesis and tumour suppression, DNA repair, inflammation, oxidation and apoptosis). Genotyping was performed using TaqMan probes or restriction fragment length assays in peripheral blood DNA. In addition, 259 paired controls were also evaluated with the same risk factors for each specific location. Nine SNPs in DNA repair (*ERCC1* rs11615, *ERCC2* rs13181), inflammatory (*IL2* rs2069762, *IL6* rs1800795), oxidative (*NFE2L2* rs13035806 and rs2706110) and apoptotic genes (*TP53* rs1042522, *MDM2* rs2279744, *BCL2* rs2279115) were differently associated with HNSCC susceptibility by location. Some of these SNPs were not described before in this tumour type. In conclusion, we describe several SNPs associated with HNC in a Spanish population.

## 1. Introduction 

Head and neck cancer (HNC) includes a set of diverse neoplasms located in the lips, oral cavity, pharynx, larynx, salivary glands and thyroid glands, among others. Most HNC belong to the squamous cell carcinomas group [1]. Approximately 600,000 new cases are diagnosed per year, being the sixth cancer type on incidence worldwide. Treatment of early stages includes surgery and/or radiotherapy, while locally advanced tumours are also treated with chemotherapy and biological therapies [2]. Only 40–50% of patients survive for five years [3] causing an annual death rate of 271,000 patients [4,5]. 

The oncogenic transformation of normal mucosa into a squamous cell carcinoma of the head and neck is a multifactorial process, associated with a variety of risk factors. At least 75% of head and neck squamous cell carcinomas (HNSCC) are attributable to the combination of cigarette smoking and alcohol drinking, the most classic carcinogens [6,7]. Diverse epidemiological studies have also revealed the existence of other environmental and genetic related factors. Similar to other tumours, viral aetiology has shown an implication in HNSCC development, predominating Epstein–Barr virus (EBV) infection in nasopharynx, and human papillomavirus (HPV), mainly subtype 16, in oral cavity and oropharyngeal tumours [8]. The carcinogenesis process triggered by viral infection defines a different entity to that caused by tobacco and alcohol [8], allowing HNSCC classification into two main prognostic and therapeutic groups, in which HPV-negative tumours are associated with an aggressive course and a worse prognosis than HPV positive ones [9,10]. 

Despite the defined role of environmental factors, there is also evidence of familial aggregation and increased cancer risk amongst HNSCC relatives [11], suggesting the existence of genetic predisposition factors [12]. However, not all individuals exposed to these carcinogens will develop the disease. In this context, the identification of genetic variants in important signaling pathways could help to define tumour susceptibility, as well as differences in treatment response and toxicity. HNSCC carcinogenesis involves different pathways: carcinogen metabolism, DNA repair, cell cycle, immunity and inflammation [13,14,15]. Single nucleotide polymorphism (SNP) is the most abundant form of genetic variation, becoming an ideal genetic susceptibility marker [1]. 

In this study, we aimed to examine polymorphisms in genes involved in relevant oncogenic pathways within a paired population of cases and controls in a large Spanish population. 

## 2. Results 

### 2.1. Characteristics of Groups 

After the application of the propensity score method 126 larynx, 100 pharynx and 70 oral cavity squamous cell carcinomas were totally paired with their specific control group. The analysis by location did not show any statistically significant difference between sex, age, tobacco and alcohol intake with respect to the control group (Table 1). Only age was statistically different (*p* < 0.05) between laryngeal tumour and control group, so this variable was included in the logistic regression as an adjustment variable. 

### 2.2. Candidate Gene Association Study

Nine out of twenty-nine selected SNPs showed a statistically significant result in the distribution between the patient and control groups. 

Beginning with DNA repair genes, less common genotypes in *ERCC1* rs11615 (*p* = 0.011, OR = 0.288 (CI 95% = 0.110–0.751) in a recessive model) and *ERCC2* rs13181 (*p* = 0.046, OR = 0.375 (0.143–0.982) in a codominant model) were associated with a lower risk of laryngeal cancer (Table 2 and Appendix A).

Secondly, pro-inflammatory *IL6* rs1800795 polymorphism was related to a higher risk of laryngeal cancer in a dominant model (*p* = 0.002, OR = 2.394 (1.376–4.163)) (Table 2 and Appendix A), similar to the association found in CG+GG variants with increased oral cavity susceptibility (*p* = 0.018, OR = 2.265 (1.148–4.467)). Moreover, another SNP in the inflammatory gene *IL2* rs2069762 G variant was associated with a lower risk of oral cavity cancer (GG *p* = 0.039, OR = 0.300 (0.096–0.940)) (Table 3 and Appendix A).

In relation to apoptotic genes, three SNPs in apoptotic genes were associated with different susceptibility in all HNSCC locations. The *TP53* rs1042522 mutant allele in the recessive model was associated with a decreased risk of developing laryngeal cancer (*p* = 0.002, OR = 0.286 (0.119–0.607)) (see Table 2 and Appendix A); and pharyngeal cancer (*p* = 0.001, OR = 0.124 (0.035–0.476)) (see Table 4 and Appendix A). Further, variant allele in *MDM2* rs2279744 was associated with higher risk of laryngeal cancer (*p* = 0.029 OR = 2.413 (1.094–5.323)) in a recessive model (Table 2 and Appendix A). Meanwhile CA+AA genotypes in *BCL2* rs2279115 were related with a higher risk of developing oral carcinoma (*p* = 0.010, OR = 2.753 (1.273–5.952)) in a dominant model (Table 3 and Appendix A). 

Finally, an association between antioxidative SNPs and laryngeal and pharyngeal cancer was found. Variant genotypes rs1303586 GA+AA and rs2706110 CT+TT, both in the *NRF2* gene, were associated with a lower risk of laryngeal carcinoma (*p* = 0.035, OR = 0.478 (0.240–0.949) and *p* = 0.518, OR = 0.518 (0.299–0.900), respectively) (Table 2 and Appendix A). On the other hand, in pharyngeal cancer, only *NRF2* rs2706110 less common allele genotypes CC+CT were related with a lower risk of developing pharyngeal carcinoma (*p* = 0.043, OR = 0.552 (0.311–0.982)) (Table 4 and Appendix A).

## 3. Discussion 

Not all individuals exposed to the same classical carcinogens (tobacco and alcohol) develop HNSCC. Although several susceptibility studies have identified SNPs in carcinogenesis-related pathways, their results are controversial due to an inadequate control group. In this multicentre case-control study, we examined the association between some polymorphisms and HNSCC susceptibility in a Spanish cohort with a control group totally paired by their risk factors, avoiding confounder variables. 

Analysis of laryngeal squamous cell carcinoma showed an association with lower susceptibility risk in *ERCC1* rs11615 and *ERCC2* rs13181 SNPs. Indeed, these genotypes have also been associated with a better response and longer survival in patients treated with platinum [16] due to an increase in DNA damage and induction of cell death, providing a potential explanation of our results.

Inflammation has been considered an important factor in the pathogenesis of human cancer [17,18,19], with a special interest in the context of oral cancer [20,21]. The rs1800795 -174C variant in the promoter of the *IL6* gene is related to a lower level of serum proteins, while -174G corresponds to a higher expression, increasing the inflammatory response [22]. Our study shows an association between the G allele and a higher risk of developing laryngeal and oral tumours, probably related to the carcinogenesis induced by inflammation. Moreover, cytokine *IL-2* plays a role in the proliferation of activated T-lymphocytes and in the activation of phagocytes. The G allele in the -330G>T (rs2069762) SNP increases the *IL2* gene expression, whereas the T allele is associated with a decreased *IL2* expression skewing the Th1/Th2 immune balance towards Th2 [23]. In our study, the *IL2* rs2069762 GG genotype was associated with lower oral cavity risk, in contrast to previous reported associations [23] in another tumour types with different risk factors and ethnic background. This result could be explained by the main role of IL-2 in the elimination of self-reactive cells [24], decreasing the antitumour response produced by the immune system. 

Mdm2 attenuates the tumour suppressor protein p53 through proteasomal degradation via ubiquitinylation, while p53 induces *MDM2* transcription in response to genotoxic stress [25]. SNP rs2279744 -410T>G, located in the P2 promoter, increases *MDM2* expression by improving the binding affinity with the Sp1 transcription factor, attenuating the *TP53* suppressor pathway [26]. Our data is in line with previous reports [27], demonstrating a higher risk of laryngeal cancer in those patients with the GG genotype.

The polymorphism c.215C>G (Pro72Arg) in the exon 4 of *TP53* is found in an essential domain in the apoptotic response and carcinogenesis inhibition. The arginine allele is a more powerful apoptotic inductor than the proline one [28,29]. Some studies have associated the Pro72Arg polymorphism with an increased risk of developing gastric, oesophageal and bladder cancer [30,31], but little data has been reported regarding HNSCC [32,33]. Our results show a lower susceptibility of developing pharyngeal and laryngeal cancer for the variant alleles. While this could be due to its association with longer survival or modifications at cell cycle and the maintenance of DNA integrity [28,34], we have also shown this protective association in stroke [35] and other ischemic processes (Cruz-González et al., data not published), possibly being a case selection bias. 

In addition, we found a statistically significant association between the anti-apoptotic gene BCL2 SNP and oral cavity cancer susceptibility. *BCL2*-938C>A (rs2279115) polymorphism is found in P2 gene promoter, acting as a negative regulator element, decreasing P1 promoter activity [36]. The presence of C allele highly reduces the activity of P1 and Bcl-2 protein expression, increasing apoptosis. Our results showed similar results to those reported in breast cancer and acute myeloid leukaemia [37] where the presence of the A allele (CA+AA) increased tumour susceptibility due to an anti-apoptotic effect [38]. 

Finally, *NFE2L2* gene codes for a transcription factor protein (Nrf2) that induce many antioxidative genes under oxidative stress. SNPs in this gene have been associated with cancer risk [39]. In our series, *NFE2L2* rs2706110 and rs1303586 less common genotypes were linked with lower risk of developing laryngeal cancer, while in pharyngeal cancer, only rs1303586 was associated. Functional analyses of these SNPs have not yet been described but our hypothesis is that these changes could increase antioxidative gene induction under stress produced at high levels in HNSCC by tobacco and alcohol consumption. 

## 4. Material and Methods

### 4.1. Study Population 

TTCC-2010-05 was an observational multicentre study conducted in 19 Spanish centres, all of them belonging to the Spanish Head and Neck Cancer Treatment Group (TTCC) coordinated by the Medical Oncology Department of the University Hospital of Salamanca, between January 2012 and December 2014. Epidemiological and clinicopathological details have been previously described [40]. 

Cases inclusion criterion was: histologically confirmed HPV-negative HNSCC patients from larynx, oro/hypopharynx and oral cavity carcinomas. They were recruited in Oncology, Radiotherapy and Otorhinolaryngology departments. Controls were follow-up individuals with minor issues and without a tumour history and paired by age, sex, smoking and alcoholic habit with HNSCC cases. They were captured in Pneumology, Radiotherapy, Otorhinolaryngology and Internal Medicine departments. Only the Spanish population were permitted, avoiding ethnicity bias. 

Considering HNSCC incidence in Spain, 10% of possible losses and duration of the study, initial calculations of recruitment were of 440 individuals in patient and control group. Finally, a total of 459 patients and 259 controls were included. 

In this study, the variables were polymorphisms in oncogenes, tumour suppressor genes, genes implicated in DNA reparation, inflammation, carcinogen metabolism and apoptosis, together with some risk factors collected in the socio-demographic (6 questions) and the data informed by patients (19 questions) questionnaires. The information of both questionnaires was collected via auto-application, being supervised by the members of the research team. Clinicopathologic data, response and specific toxicity to treatment were collected by oncologists in the case report form questionnaire (CRF). 

The study was approved by the University Hospital of Salamanca and the local ethics committees in accordance with the 1964 Helsinki declaration and its later amendments. All participants were previously informed and signed the provided informed consent. All data were treated with the security measures established in compliance with the Protection of Personal Data Organic Law 15/1999, December 13, and safe-keeping by the University Hospital of Salamanca in its specific hospital server. This study was supported by the Ministry of Economy and Competitiveness under the identification code PI11/0059.

### 4.2. Selection of Polymorphism 

Candidate SNPs selection was done according to at least two of the following criteria: >5% allele frequency in Caucasian/European population, previously defined association with HNSCC susceptibility and earlier related different response or toxicity to chemotherapy or radiotherapy. At the initial stages of the project design, a huge search was performed in available databases using keywords such as SNPs, susceptibility, HNSCC, response and toxicity, selecting only those with statistically significant results in other populations [14,15,32,41,42,43]. SNPs with some published evidence of functionality were preferably selected Table 5. 

### 4.3. DNA Isolation and Genotyping 

DNA was extracted from peripheral blood leukocytes using the phenol-chloroform method. Genotyping was performed using the TaqMan^®^ Allelic Discrimination Assay [44] (Applied Biosystems, Foster, CA, USA) in those SNPs where the probes were available. A concentration of 40 ng/μL of DNA samples were added to 6.25 μL of Taqman^®^ Universal PCR Master Mix and it was combined with specific forward and reverse primers, and allele-specific VIC (allele 1) and FAM (allele 2) labelled probes. The assay was performed in a 96 well plate and the detection was measured in the Step One Plus Real-Time PCR System Thermal Cycling Block (Applied Biosystems, Foster, CA, USA). Negative and positive controls were always added. A total of 5% of random samples were re-genotyped to ensure the reproducibility. 

In those candidate SNPs in which TaqMan® probes were not available, genotyping was analysed using polymerase chain reaction—restriction fragment length polymorphism (PCR-RFLP). Specific oligonucleotides were designed to amplify the polymorphic sequences and digestion was made via the specific restriction enzymes. The PCR products were run on 3% Syber-safe stained agarose gel and visualized under UV light. GSTT1 and GSTM1 null/present SNPs were analysed using PCR with β-actin as an endogenous control. Finally, for KRAS-LC6 rs61764370, a custom probe was specifically designed. Sequences and type of assays are shown in Appendix A.

### 4.4. Statistical Analysis 

The statistical analysis to associate the relation between the different clinical and molecular variables was analysed using cross tabs and the χ^2^ test of Pearson. The odds ratios (OR) and 95% confidence intervals were calculated using logistic regression analysis. The quantitative variable distribution was analysed using the ANOVA test in those examples where the sample followed a parametric distribution (*p* > 0.05 in Levene’s test), while in those with a non-parametric distribution, a Mann–Whitney *U* test was applied. Hardy–Weinberg equilibrium (HWE) was tested in a control population using a χ^2^ test. Statistically significant differences were considered to exist when the two-sided *p*-value was <0.05. Only TP53 rs1042522 and APEX rs1130409 were in disequilibrium (*p*HWE < 0.05). 

Because of the lower inclusion of matched controls, the statistical analysis was realised when matching the group via the propensity score method (PS). This allowed us to equate groups in a cohort study through a logistic regression introducing the confounders as predictive variables [45,46,47]. Groups were matched according to: packs per year consumed (PPY): no smokers, <20PPY and >20PPY, standard unit of alcohol per week (SDU/week): <14SDU/week and >14SDU/week, and sex. Quantitative age was not included in the PS and it was introduced in the logistic regression as adjustment variable only in laryngeal cancer where the age between both groups was statistically significant. 

These analyses were performed with the statistical software SPSS v.21.0 (IBM-SPSS Inc., Chicago, IL, USA). 

## 5. Conclusions

This study shows the association between several polymorphisms in genes involved in DNA repair, inflammation, antioxidative and apoptotic pathways with susceptibility to developing HPV-negative HNSCC. The characteristics of the control group positively indicates that these results are caused by the genetic background, avoiding confounder variables. Likewise, the differences found in this association study according to the location corroborate the heterogeneity in these tumours included under the same term of head and neck squamous cell carcinoma. It is important to mention that this study could provide evidence to define the consideration of different genetic entities within HNSCC and the necessity of using a matched control population by their risk factors in future case-control studies. Larger studies should be performed and would be necessary to confirm these results. 

## Figures and Tables

**Table 1 cancers-11-00493-t001:** Descriptive characteristics and risk factors of paired patients by location in the case-control study. Data after the propensity score method corroborate the equality between the different locations with their specific controls, except for age in laryngeal tumours (introduced as an adjustment variable in the logistic regression).

Group Comparison	LARYNX *N* = 126		CONTROL *N* = 126		*p*-Value	PHARYNX *N* = 100		CONTROL *N* = 100		*p*-Value	ORAL CAVITY *N* = 70		CONTROL *N* = 70		*p*-Value
**Characteristics**	*N*	**%**	*N*	**%**		*N*	%	*N*	%		*N*	%	*N*	%	
**Age (years)**	63.02 ± 8.566		56.30 ± 12.803		**0.000**	59.96 ± 8.41		59.52 ± 10.044		0.742	60.92 ± 10.008		62.24 ± 8.88		0.412
**Sex**															
**Female**	13	10.3	13	10.3	1.000	20	20.0	22	22.0	0.728	16	22.9	17	22.9	1.000
**Male**	113	89.7	113	89.7		80	80.0	78	78.0		54	77.1	54	77.1	
**Tobacco smoking**															
**Never**	7	5.5	7	5.5	0.944	7	7.0	8	8.0	0.943	7	10.0	7	10.0	1.000
**<20 PPY**	20	15.9	22	17.5		22	22.0	23	23.0		12	17.1	12	17.1	
**>20 PPY**	99	78.6	97	77.0		71	71.0	69	69.0		51	72.9	51	72.9	
**Missing**	0	0	0	0		0	0	0	0		0	0	0	0	
**Alcohol drinking**															
**Never**	53	42.1	51	40.5	0.904	26	26.0	27	27.0	0.985	23	32.9	23	32.9	1.000
**<14 SDU/week**	28	22.2	31	24.6		30	30.0	30	30.0		19	27.1	19	27.1	
**>14 SDU/week**	45	35.7	44	34.9		44	44.0	43	43.0		28	40.0	28	40.0	
**Missing**	0	0	0	0		0	0	0	0		0	0	0	0	

*p*-values related to controls. Statistically significant results in bold. PPY: Tobacco packs per year. SDU: Standard unit of alcohol per week.

**Table 2 cancers-11-00493-t002:** Statistically significant SNPs in laryngeal cancer.

SNPs	Genotype	Larynx		Control		*p*-Value *	OR (CI 95%)
		*N*	%	*N*	%		
***TP53* rs1042522**	GG	61	48.4	62	49.2	Ref.	1.00
	GC	54	42.9	37	29.4	0.165	1.505 (0.846–2.677)
	CC	11	8.7	27	21.4	**0.008**	**0.319 (0.136–0.745)**
Recessive	GG+GC	115	91.3	99	78.6	Ref.	1.00
	CC	11	8.7	27	21.4	**0.002**	**0.268 (0.119–0.607)**
Dominant	GG	61	48.4	62	49.2	Ref.	1.00
	GC+CC	65	51.6	64	50.8	0.596	0.986 (0.587–1.654)
***MDM2* rs2279744**	TT	44	34.9	62	49.2	Ref.	1.00
	TG	57	45.2	53	42.1	0.279	1.364 (0.778–2.392)
	GG	25	19.8	11	8.7	**0.015**	**2.826 (1.219–6.552)**
Recessive	TT+TG	101	80.2	115	91.3	Ref.	1.00
	GG	25	19.8	11	8.7	**0.029**	2.413 (1.094–5.323)
Dominant	TT	44	34.9	62	49.2	Ref.	1.00
	TG+GG	82	65.1	64	50.8	0.075	1.616 (0.953–2.742)
***ERCC1* rs11615**	TT	53	42.1	45	35.7	Ref.	1.00
	TC	67	53.2	58	46.0	0.872	0.956 (0.550–1.661)
	CC	6	4.8	23	18.3	**0.013**	**0.281 (0.103–0.768)**
Recessive	TT+TC	120	95.2	103	81.7	Ref.	1.00
	CC	6	4.8	23	18.3	**0.011**	**0.288 (0.110–0.751)**
Dominant	TT	53	42.1	45	35.7	Ref.	1.00
	TC+CC	73	57.9	81	64.3	0.354	0.778 (0.457–1.324)
***ERCC2* rs13181**	TT	72	57.1	52	41.3	Ref.	1.00
	TG	46	36.5	58	46.0	0.247	0.720 (0.413–1.255)
	GG	8	6.3	16	12.7	**0.046**	**0.375 (0.143–0.982)**
Recessive	TT+TG	118	93.7	110	87.3	Ref.	1.00
	GG	8	6.3	16	12.7	0.079	0.433 (0.170–1.102)
Dominant	TT	72	57.1	52	41.3	Ref.	1.00
	TG+GG	54	42.9	74	58.7	0.093	0.638 (0.377–1.078)
***IL6* rs1800795**	CC	43	34.1	62	50.8	Ref.	1.00
	CG	64	50.8	46	37.7	**0.003**	**2.471 (1.372–4.452)**
	GG	19	15.1	14	11.5	0.070	2.164 (0.938–4.991)
Recessive	CC+CG	107	84.9	108	88.5	Ref.	1.00
	GG	19	15.1	14	11.5	0.444	1.351 (0.625–2.921)
Dominant	CC	43	34.1	62	50.8	Ref.	1.00
	CG+GG	83	65.9	60	49.2	**0.002**	**2.394 (1.376–4.163)**
***NRF2* rs1303586**	GG	109	87.2	95	76.0	Ref.	1.00
	GA	14	11.2	29	23.2	**0.019**	**0.424 (0.207–0.869)**
	AA	2	1.6	1	0.8	0.520	2.235 (0.193–25.903)
Recessive	GG+GA	123	98.4	124	99.2	Ref.	1.00
	AA	2	1.6	1	0.8	0.444	2.600 (0.225–30.064)
Dominant	GG	109	87.2	95	76.0	Ref.	1.00
	GA+AA	16	12.8	30	24.0	**0.035**	**0.478 (0.240–0.949)**
***NRF2* rs2706110**	CC	92	73.6	72	57.1	Ref.	1.00
	CT	24	19.2	47	37.3	**0.005**	**0.425 (0.233–0.775)**
	TT	9	7.2	7	5.6	0.732	1.207 (0.411–3.541)
Recessive	CC+CT	116	92.8	119	94.4	Ref.	1.00
	TT	9	7.2	7	5.6	0.403	1.574 (0.544–4.560)
Dominant	CC	92	73.6	72	57.1	Ref.	1.00
	CT+TT	33	26.4	54	42.9	**0.020**	**0.518 (0.299–0.900)**

* *p*-values adjusted by age. Statistically significant results in bold.

**Table 3 cancers-11-00493-t003:** Statistically significant SNPs in oral cavity carcinoma.

SNPs	Genotype	Oral Cavity		Control		*p*-Value	OR (CI 95%)
		*N*	%	*N*	%		
***IL2* rs2069762**	**TT**	**43**	61.4	31	44.3	/	1.00
	TG	22	31.4	27	38.6	0.152	0.587 (0.284–1.217)
	GG	5	7.1	12	17.1	**0.039**	**0.300 (0.096–0.940)**
Recessive	TT+TG	65	92.9	58	82.9	/	1.00
	GG	5	7.1	12	17.1	0.078	0.372 (0.124–1.119)
Dominant	TT	43	61.4	31	44.3	/	1.00
	TG+GG	27	38.6	39	55.7	**0.043**	**0.499 (0.254–0.979)**
***IL6* rs1800795**	CC	25	35.7	39	55.7	/	1.00
	CG	33	47.1	23	32.9	**0.031**	**2.238 (1.077–4.653)**
	GG	12	17.1	8	11.4	0.104	2.340 (0.839–6.528)
Recessive	CC+CG	58	82.9	62	88.6	/	1.00
	GG	12	17.1	8	11.4	0.337	1.603 (0.612–4.203)
Dominant	CC	25	35.7	39	55.7	/	1.00
	CG+GG	45	64.3	31	44.3	**0.018**	**2.265 (1.148–4.467)**
***BCL2* rs2279115**	CC	13	18.6	27	38.6	/	1.00
	CA	43	61.4	30	42.9	**0.008**	**2.977 (1.325–6.688)**
	AA	14	20.0	13	18.6	0.116	2.237 (0.820–6.103)
Recessive	CC+CA	56	80.0	57	81.4	/	1.00
	AA	14	20.0	13	18.6	0.830	1.096 (0.473–2.540)
Dominant	CC	13	18.6	27	38.6	/	1.00
	CA+AA	57	81.4	43	61.4	**0.010**	**2.753 (1.273–5.952)**

Statistically significant results in bold.

**Table 4 cancers-11-00493-t004:** Statistically significant SNPs in pharyngeal cancer.

SNPs	Genotype	Pharynx		Control		*p*-Value	OR (CI 95%)
		*N*	%	*N*	%		
***TP53* rs1042522**	GG	53	53.0	47	47.0	Ref.	1.00
	GC	44	44.0	33	33.0	0.583	1.182 (0.650–2.151)
	CC	3	3.0	20	20.0	**0.002**	**0.133 (0.037–0.476)**
Recessive	GG+GC	97	97.0	80	80.0	Ref.	1.00
	CC	3	3.0	20	20.0	**0.001**	**0.124 (0.035–0.431)**
Dominant	GG	53	53.0	47	47.0	Ref.	1.00
	GC+CC	47	47.0	53	53.0	0.396	0.786 (0.451–1.370)
***NRF2* rs2706110**	CC	68	68.0	54	54.0	Ref.	1.00
	CT	25	25.0	41	41.0	**0.020**	**0.484 (0.262–0.893)**
	TT	7	7.0	5	5.0	0.863	1.112 (0.334–3.698)
Recessive	CC+CT	93	93.0	95	95.0	Ref.	1.00
	TT	7	7.0	5	5.0	0.553	1.430 (0.438–4.667)
Dominant	CC	68	68.0	54	54.0	Ref.	1.00
	CT+TT	32	32.0	46	46.0	**0.043**	**0.552 (0.311–0.982)**

Statistically significant results in bold.

**Table 5 cancers-11-00493-t005:** SNPs selected in the study. Candidate SNPs were selected in oncogenes and tumour suppressor genes, DNA repair (either BER, NER and DSB), inflammatory, apoptotic and carcinogen metabolism genes, as described in Material and Methods.

FUNCTION	GENE	RS	ID	Change
Oncogenes and tumour suppressor genes	*TP53*	1042522	C_2403545_10	Pro72Arg
	*MDM2*	2279744	PCR-RFLP	-410T>G
	*KRAS-LC6*	61764370	PCR-custom probe	3’-UTR
	*EGFR*	2227983	C_16170352_20	Lys521Arg
Base excision repair (BER)	*XRCC1*	25487	C_622564_10	Gln399Arg
		1799782	C_11463404_10	Arg194Trp
	*APEX*	1130409	C_8921503_10	Asp148Glu
Nucleotide excision repair (NER)	*ERCC2(XPD)*	13181	C_3145033_10	Lys751Gln
	*ERCC1*	11615	C_2532959_10	Asn118Asn
	*XPC*	2228000	C_16018061_10	Ala499Val
Double-strand break repair genes (DSB)	*XRCC3*	861539	C_8901525_10	Thr241Met
		1799794	C_2983904_10	-316A>G
	*KU70*	2267437	C_15872242_20	-731C>G
Inflammatory genes	*IL1B*	16944	C_1839943_10	-511T>C
	*IL2*	2069762	C_15859930_10	-330T>G
	*IL6*	1800795	C_1839697_20	-174C>G
	*IL10*	1800872	C_1747363_10	-592C>A
	*TNFA*	361525	C_2215707_10	-238A>C
Apoptotic genes	*NOD2*	2066844	C_11717468_20	Arg702Trp
		2066845	C_11717466_20	Arg908Gly
	*BAX*	4645878	C_27848291_10	-248G>A
	*BCL2*	2279115	C_3044428_30	-938C>A
Carcinogen metabolism/antioxidative genes	*CYP3A5*	776746	C_26201809_30	6986A>G
	*GSTP1*	1695	C_3237198_20	Ile105Val
	*GSTT1*	N/A	PCR	Null/present
	*GSTM1*	N/A	PCR	Null/present
	*NFE2L2* *(NRF2)*	13035806	C_11745134_10	3’-UTR
		2706110	C_11745133_10	3’-UTR
	*KEAP1*	1048290	C_9323035_10	Leu471Leu

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
