# Peer review of "Genetic Susceptibility in Head and Neck Squamous Cell Carcinoma in a Spanish Population"

_cancers, 2019, doi:10.3390/cancers11040493_

Reviewer 1 Report

The study is an well performed analysis of a set of SnPs in Spanish HNSCC patients, and is performed with appropriate statistical analysis. 

Increasing the analysis the independence of the significant SNPs would improve the quality of the manuscript. 

The authors should include some bioinformatics analysis to help to define the functional impact of the SNPs on protein function, and the overall impact of the study would be hugely increased with a supporting functional study comparing protein function with and without the significant SNPs. 

Author Response

The study is a well performed analysis of a set of SNPs in Spanish HNSCC patients, and is performed with appropriate statistical analysis. 

Increasing the analysis, the independence of the significant SNPs would improve the quality of the manuscript. 

As we cited in the last paragraph of the discussion (line 172), we are concerned that the main limitation of our study is the sample size. However, one of the most important objectives was to match the patients and controls according to tobacco and alcohol consumption, sex and age. For that purpose, Propensity Score mathematical method was applied. Although the total number of patients recruited was 459, after Propensity Score application, they were reduced to 296; totally paired with controls in terms of risk factors. Unfortunately, we could not include more individuals who fulfil all the study characteristics, but we think that the results obtained in this manuscript provide clear evidence that can be validated in further case-control studies.  

The authors should include some bioinformatics analysis to help to define the functional impact of the SNPs on protein function, and the overall impact of the study would be hugely increased with a supporting functional study comparing protein function with and without the significant SNPs. 

Selection criteria is defined in line number 203. At the beginning of the study, SNP selection was done with previously described and characterized polymorphism, published in papers and different databases such as NCBI, COSMIC and SNPedia among others, selecting preferably those ones with published evidence of functionality (as it is cited in lines 208-209). For that reason, we did not include any bioinformatic analysis.

Reviewer 2 Report

Reviewer's report

Title

Genetic susceptibility in head and neck squamous cell carcinoma in a Spanish population

Version: 1 Date:  2019 March 10th

Reviewer's report:

In this manuscript, the Authors evaluated a list of 29 SNPs, previously reported to be associated with HNSCC, in a series of 296 HPV negative HNSCC and 259 paired controls from a Spanish population. They found a total of 9 SNPs in DNA repair, inflammatory, oxidative and apoptotic genes to be associated to HNSCC susceptibility.

Interestingly, the same group published in 2017 a similar paper identifying autophagy gene polymorphisms association with HNSCC. However, they did not cite it in this manuscript (see Sci Rep. 2017 Jul 31;7(1):6887.).

This work is well structured and written, however it suffered from the following points:

Major points:

Section Material and Methods – page 8, lane 223: please specify primer sequences,  restriction enzymes used and temperature in a new Table.

Section Material and Methods – page 7 – lane 182: Since controls come from hospitalized patients without tumour history and captured in Pneumology, Radiotherapy, Otorhinolaryngology and Internal Medicine departments, a bias due to a possible association between their diseases (inflammatory, degenerative, hyperproliferative, or whatever) and the list of 29 SNPs must be taken in consideration and discussed. 

Section Material and Methods – page 7 – lane 187: Please check the real number of patients included, as in the abstract the number of patients involved was 296 and 259 controls; while the Authors wrote “ Finally, a total of 459 patients and 259 controls were included.”

Minor points: 

-      Gene name must be written in italics

Author Response

In this manuscript, the Authors evaluated a list of 29 SNPs, previously reported to be associated with HNSCC, in a series of 296 HPV negative HNSCC and 259 paired controls from a Spanish population. They found a total of 9 SNPs in DNA repair, inflammatory, oxidative and apoptotic genes to be associated to HNSCC susceptibility.

Interestingly, the same group published in 2017 a similar paper identifying autophagy gene polymorphisms association with HNSCC. However, they did not cite it in this manuscript (see Sci Rep. 2017 Jul 31;7(1):6887.).

This work is well structured and written, however it suffered from the following points:

 Major points:

Section Material and Methods – page 8, lane 223: please specify primer sequences, restriction enzymes used and temperature in a new Table.

We have modified this in the manuscript. Table with specific primer sequences and restriction enzymes are displayed in Supplementary table S1. Apart from that, we modified in Table 1 KRAS-LC6 rs 61764370 ID because this SNP was done by PCR and custom designed TaqMan® probe instead of PCR-RFLP. Better description of the assays was described in Material and methods (lines 234-236): GSTT1 and GSTM1 null/present SNPs were analysed by PCR with β-actin as an endogenous control. Finally, for KRAS-LC6 rs61764370, a custom probed was specifically designed. Sequences and type of assays are shown in Table S1.

Table S1. Primer sequences and genotyping assay in those SNPs defined in Table 1 without specific TaqMan® probes.

Primer

Primer sequence   (5’-3’)

Type of assay

MDM2

rs2279744

forward

5’-CAGGTCTCCGCGGGAGTTC-3’

Digestion with MspA1I

reverse

5’-CGTGTCTGAACTTGACCAGC-3’

KRAS-LC6

rs61764370

forward

5’-GCCAGGCTGGTCTCGAA-3’

Custom probe

CTCAAGTGAT[T/G]CACCAC

reverse

5’-CTGAATAAATGAGTTCTGCAAAACAGGTT-3’

GSTT1

forward

5’-TTCCTTACTGGTCCTCACATCCTC-3’

Null/present

reverse

5’-TCACCGGATCATGGCCAGCA-3’

GSTM1

forward

5’CGCCATCTTGTGCTACATTGCCCG-3’

reverse

5’-TTCTGGATTGTAGCAGATCA-3’

β-actin

forward

5’-CCAACCGCGAGAAGATGA-3’

reverse

5’-CCCGAGGCGTACAGGGATAG-3’

Section Material and Methods – page 7 – lane 182: Since controls come from hospitalized patients without tumour history and captured in Pneumology, Radiotherapy, Otorhinolaryngology and Internal Medicine departments, a bias due to a possible association between their diseases (inflammatory, degenerative, hyperproliferative, or whatever) and the list of 29 SNPs must be taken in consideration and discussed. 

This is a very interesting observation. We should not have used the term hospitalized because it can be confusing. They were follow-up individuals with minor issues who came to the different departments described above. For instance, most of the controls recruited at the Pneumology Department were individuals going to the tobacco withdrawal Unit, those in Internal Medicine Department were part of the Alcoholics anonymous Unit, and so on. To address this limitation, we have changed the term hospitalized patients for follow-up individuals with minor issues in lines 187-188.

However, given the well characterized association of these SNPs with oncogenic molecular processes it is unlikely that they could be associated with other diseases. In addition, all clinical antecedents were recorded in the questionnaires avoiding the selection of individuals with serious illnesses as controls. 

Section Material and Methods – page 7 – lane 187: Please check the real number of patients included, as in the abstract the number of patients involved was 296 and 259 controls; while the Authors wrote “ Finally, a total of 459 patients and 259 controls were included.”

Sample size is a key point in this study. Originally, following inclusion criteria, 459 patients and 259 controls were recruited (as cited in line 187). However, after pairing both groups according to their matching characteristics by Propensity Score, 296 patients were definitely used for the association study (as cited in the abstract-line 24 and results-line 68). The rest of the patients were excluded because they did not present the same risk factors as controls.

Minor points: 

-      Gene name must be written in italics

We have modified this accordingly.

Round  2

Reviewer 1 Report

The authors did not make significant changes to the manuscript in response to the comments, and did not interpret how the SNPs may impact biological function, which is disappointing. However, the overall study is relatively well performed on the cohort. 

Author Response

Please find attached the reviewer comments

Reviewer 2 Report

Dear Editor,

The Authors provided appropriate revisions for all of the issues I suggested. Therefore, I can recommend this paper for publication on Cancers.

Author Response

Plase find attached the reviewer coments
